# “Who Cares?”: The Acceptance of Decentralized Wastewater Systems in Regions without Water Problems

**DOI:** 10.3390/ijerph17239060

**Published:** 2020-12-04

**Authors:** Cristina Gómez-Román, Luisa Lima, Sergio Vila-Tojo, Andrea Correa-Chica, Juan Lema, José-Manuel Sabucedo

**Affiliations:** 1CRETUS Institute and Department of Social Psychology, Basic and Methodology, Universidade de Santiago de Compostela, 15705 Santiago de Compostela, Spain; cristina.gomez@usc.es (C.G.-R.); sergiovila.tojo@usc.es (S.V.-T.); 2CIS-IUL/ISCTE, Instituto Universitário de Lisboa, 1649-026 Lisboa, Portugal; luisa.lima@iscte-iul.pt; 3Department of Social Psychology, Basic and Methodology, Universidade de Santiago de Compostela, 15705 Santiago de Compostela, Spain; jullyandrea.correa@rai.usc.es; 4CRETUS Institute and Department of Chemical Engineering, Universidade de Santiago de Compostela, 15705 Santiago de Compostela, Spain; juan.lema@usc.es

**Keywords:** decentralized plants, systematic review, focus group, public acceptance

## Abstract

There is a growing interest in decentralized wastewater treatment systems, especially in regions with water scarcity problems or water management issues. This study aims to determine whether the perceived advantages and disadvantages (leading to acceptance) of decentralized wastewater plants in such regions are the same in regions where the population is not aware of these water issues. Firstly, this study systematically reviews previous findings on public perceptions of the acceptance of decentralized wastewater treatment systems. Then, the study details the results of a focus group study to determine whether the elements identified in the literature are also relevant in a region where people are unaware of water problems. The results show that a lack of awareness of water issues seems to be a critical factor influencing acceptance. Reframing the usefulness of these systems by focusing on other aspects, such as environmental sustainability, is key.

## 1. Introduction

Beginning in the second half of the 19th century, piped water supplies in urban areas has been based on the hydraulic paradigm [1,2]. Municipal, regional, and state governments take the lead in planning and constructing large-scale water and wastewater infrastructure [3]. Only at smaller scales can infrastructure be managed by private firms or through local collective action.

The current water crisis related to water scarcity and environmental impacts [4], and the limitations of the hydraulic paradigm to solve this crisis [5], are leading to a change in water management approaches [6]. To ensure the sustainability of water treatment systems, management strategies must take a multidisciplinary approach. Such an approach should ideally integrate social, economic, and environmental aspects with practices such as rainwater management, water conservation, wastewater reuse, rational energy management, nutrient recovery, and sorting at the source.

The current debate on urban planning recognizes the importance of local solutions in the search for sustainable development [3]. The shift to decentralized management plans is now widely seen as the future of urban water management. As Rittmann (2013) points out, decentralized plants have re-emerged as possible solutions to this problem [7]. These plants allow for differentiated treatment of wastewater which has been separated from the source, enabling the recovery of water for new uses, and promoting the circular economy [8,9]. 

There are several decentralized wastewater solutions, including natural treatment systems, aerobic systems, anaerobic systems and combined (aerobic/anaerobic/natural) systems [10]. Regardless of the technology behind these approaches, however, all of them challenge the prevailing logic of the disposal of waste far from home. Instead, these procedures involve local treatment and reuse of wastewater (the use of the recovered water also depends on the system; for instance, recovered water may be used for irrigation and nutrients for fertilization, among other uses [11]). This study focuses on the concept of decentralized wastewater systems, irrespective of the type of technology or recovered products. The study examines the shared objective underlying these solutions as an alternative to current centralized wastewater schemes, which are based on the “flush and forget” philosophy [12].

Despite the potential advantages of these systems, social acceptance of these solutions is crucial to successful implementation [13,14,15,16]. A growing body of research has examined acceptance levels for these types of plants [17,18,19,20,21]. These studies reveal that people analyze the advantages and disadvantages of these systems, including the economic aspects. However, regardless of the conclusions regarding these systems’ benefits and drawbacks, an essential condition for public acceptance is that citizens must perceive that improvement of some element of the current water management is necessary. This is, of course, not always the case. 

Residents of many regions where resource management is not on the social and political agenda are satisfied with their current centralized wastewater systems. In those contexts, would people support the implementation of a decentralized system? If the solution to global problems often lies in local actions, it is necessary to identify possible barriers to the public acceptance of decentralized wastewater management systems in those regions and explain how to overcome those barriers. To achieve this objective, it is worth examining public perceptions relating to the implementation of decentralized plants in a region where people are unaware of water management issues.

This document proceeds as follows. Firstly, this paper undertakes a systematic literature review to assess and analyze the most relevant barriers and opportunities which have been reported in studies on public perceptions and the acceptance of decentralized systems. Then, this paper details the results of a focus group study that was conducted to determine whether the trends in the literature are present in the discourse of potential end users of the wastewater treatment plants.

## 2. Materials and Methods

### 2.1. Systematic Literature Review

The purpose of the literature review is to identify factors associated with public perceptions and the acceptance of decentralized wastewater treatment systems by examining research on the topic conducted around the world. The study protocol is registered in PROSPERO: International prospective register of systematic reviews (ID: CRD42018086970; [22]). This work was performed in accordance with the Preferred Reporting Items for Systematic Reviews and Meta-Analyses (PRISMA) statement [23].

#### 2.1.1. Inclusion and Exclusion Criteria

Several search terms were used to identify the relevant literature (these search terms can be found in Appendix A of the Appendix A). The authors considered materials from the following relevant electronic bibliographic databases: Academic Search Complete, PsycINFO, MEDLINE, Psychology and Behavioral Sciences Collection, PsycARTICLES, ERIC, Web of Science, Scopus, SciELO, B-On, The Cochrane Library, Open Science Directory and Google Scholar. Then, works related to human participants and environmental issues were selected, as well as work from the research areas of psychology, sociology, and social sciences. The volumes of the journals indexed in ISI and Scopus related to the topic of the systematic review were also included. In addition, peer-reviewed conferences and papers included in relevant studies in national and international government reports and non-governmental organization publications (gathered via Google search) were included. To identify other articles of interest, the authors searched the reference lists of articles, citation lists where the article was referred to, and registered protocols related to the objective of this systematic review. Reference sections of known authors were also searched using the pertinent fields on Google Scholar and ResearchGate. 

The search was restricted to English, Portuguese and Spanish articles published between January 1970 and October 2020. Publications related to the technical properties of wastewater systems and the fields of engineering and physics were excluded. 

#### 2.1.2. Selecting Appropriate Sources

Papers were downloaded from a Mendeley library. The authors reviewed all titles and abstracts of the papers to assess their relevance against predetermined inclusion and exclusion criteria. Forward and backward citation tracking supplemented the database searches. Full-text manuscripts were obtained for all studies included in the review. Any uncertainty about the inclusion of an article in the review was resolved by examining the full text of the article. 

#### 2.1.3. Data Extraction, Analysis and Synthesis 

A framework was developed to organize the literature according to comparable study contexts. This framework also facilitated synthesis of the results. For each of the selected studies, the following items were considered: topic, context, sample size, level of analysis, variables studied, data collection method and data analysis approach. Three experts in the field extracted the data directly into an Excel spreadsheet following the above-mentioned framework. Where necessary, the authors resolved uncertainties by consensus.

### 2.2. Focus Group Methodology

As mentioned above, the authors attempted to confirm whether the variables that were identified as being important in the literature review were also present in the concerns of the public in a region where the population is not aware of water issues. 

#### 2.2.1. Sample and Procedure

Galicia is a region of Spain where there is neither a problem of water scarcity (see Figure 1) nor a public debate on water management. The authors designed several discussion groups with different profiles, comprising citizens from this area. Although some publications have showed the impacts of anthropogenic global warming and pollution in this region [24], including increasing water problems related to the eutrophication of watersheds [25], loss of marine ecosystems due to major anthropogenic disturbances [26], and evidence of massive proliferation of *Microcystis aeruginosa* in water reservoirs [27], the local population is still not aware of these problems. 

Focus groups are a qualitative data collection technique that have been widely used in various areas of psychosocial research [29,30]. They are a type of interviewing technique, whereby a moderator guides an interactive discussion between a small group of individuals on specific topics. This technique is useful for obtaining detailed information about perceptions and opinions, which allows researchers to gather broader and deeper knowledge about the topic being studied. Unlike quantitative techniques, the representativeness of the data from a focus group is not gauged by the size of the sample, but by the fact that all the arguments for and against the topic under study appear in the discussion; this is known as saturation. Therefore, what is relevant in the focus group technique is not the number of people participating, but their ability to capture the various positions that may exist on a topic.

This study examined three reference groups: architects, ecologists, and the general population. The objective of this group selection was to obtain the greatest possible variability in perspectives among participants to ensure sufficient data for analysis [31]. The architects were selected for their technical expertise. Architects supervise the design of buildings and the refurbishment of old ones, often including implementation or not of these new decentralized technologies. Another important reference group this study examined was the ecologists. One cannot ignore that environmental groups have become a significant force in social and political life; their activism can both encourage and halt the implementation of new systems [32]. Additionally, two groups composed of participants from the general population were also interviewed.

A total of four focus group sessions were held in March and April 2017 with 4–5 participants at each session (18 participants in total: 11 men, 7 women; average age 41, age range 28–55). This structure (i.e., four focus groups with 4–5 participants in each group) is based on the recommendations of Krueger and Casey (1994) and Morgan (1997) [29,30]. These authors recommend that three to six different focus groups are adequate to achieve saturation (with 4–12 participants per group). Four to five participants were chosen for each group; as the number of group members increases, the likelihood that the moderator will need to intervene more than necessary rises, and some participants are more prone to inhibit their participation.

To select participants, volunteers were asked to take part in a research study on environmental concerns. The researchers contacted several social groups, such as the Technical School of Architecture of the University of A Coruña (Spain), the Association for the Ecological Defense of Galicia (ADEGA), and neighborhood associations. This allowed the researchers to identify overlaps between the three sources, which is useful evidence that contributes to the validity of the results [33].

A skilled moderator facilitated each focus group session, which lasted approximately one hour. The focus groups began by gauging participants’ current knowledge about decentralized plants, and after explaining to participants what decentralized plants are, participants were asked to identify the possible advantages and disadvantages they believed might result from the implementation of these plants. Focus group sessions were audiotaped and then transcribed.

Before each focus group discussion started, participants had to fill out a consent form, which included information about the research and the focus group process (e.g., duration, noting the discussion would be recorded, confidentiality terms). Participants were also informed that they could choose whether or not to participate and that they could withdraw from the study at any time. By signing the consent form, focus group participants voluntarily consented to be recorded and verified that they were competent to participate in the study.

#### 2.2.2. Data Analysis

The transcripts of the four focus groups were analyzed. To perform the content comparison analysis, the ATLAS.ti version 7.5 software was used, and coding was performed based on grounded theory [34]. Grounded theory is a systematic methodology that operates in an inductive manner, whereby researchers collect repeated ideas, concepts or elements that arise during the research process. Those items are then labeled with codes and can be used to generate a substantive theory based on the data.

Following this methodology, the coding process was carried out at three levels: (1) open, in which line-by-line coding of each transcript, concepts and properties of decentralized water plant technology were identified; (2) axial, in which open codes were linked according to their properties and dimensions; and (3) selective, in which the codes were integrated and refined into central categories, and the systematic relationship with other categories was identified [35]. This process allowed the identification of isolated and non-isolated incidents relevant to the construction of substantive theory. The coding process was complete when further analysis failed to uncover any new thematic ideas in relation to the emerging theory, resulting in saturation in three categories. 

During the analysis, specifically at the beginning of axial coding, it was identified that similar concepts emerged among the three reference groups (architects, ecologists, and the general population) relating to the technology of decentralized water plants. 

## 3. Results

### 3.1. Assessing Global Public Acceptance of Decentralized Wastewater Treatment Systems

The literature review conducted according to the selected terms previously mentioned identified a total of 615 publications that met the search criteria. From the above total, 267 reports were excluded because they were not directly related to the topic of the study. In the full-text review stage, the researchers identified 348 articles that comprised the existing scholarly research on public acceptance of wastewater systems. A total of 302 were excluded for a number of reasons, such as the topics of the studies being related to water recycling, biosolids or waste management (Appendix A). Finally, 46 bibliographic records were included in the review. A summary of the articles included in the review can be found in Appendix A.

The 46 bibliographic records included were fundamentally contextualized in regions with water scarcity problems of Australia, Asia, Africa, America, and Europe. Some of these studies are a compilation of previous studies, and others report results from existing decentralized plants. In the latter case, questionnaires were developed for users to assess their opinion of the plants already in operation. The following paragraphs detail the most commonly recurring variables with a summary of the most relevant results. 

#### Barriers and Facilitators for the Acceptance of Decentralized Wastewater Treatment Systems Worldwide

(a)Sociodemographic variables

Many papers have included socio-demographic details as study variables, but have not analyzed them; however, age and gender have been found to be relatively consistent predictors of the level of comfort a person feels with use of alternative water treatments [36,37,38,39]. Mankad et al. (2011) reported that women aged 25 and older were less willing to accept alternative systems than male respondents aged 18–24 [18]. They also found that as a person’s income increases, so does their willingness to accept the use of alternative water systems.

Most of the publications examined in the literature review did not find significant trends or correlations relating to public acceptance and education levels. However, in studies where these differences were significant, such as those by Lienert and Larsen (2006) and Mankad, Tucker, Tapsuwan and Greenhill (2010), the trend was that the higher an individual’s level of education, the more likely that individual is to accept the use of alternative wastewater systems [40,41].

(b)Perception of water scarcity, knowledge, and information 

Given the locations in which this type of research has mainly been conducted, the perception of water scarcity is a critical facilitating factor that conditions public attitudes towards alternative water systems. This is especially evident in countries with water restrictions such as Australia, where government reports and scientific findings suggest that without significant investments in the adoption of alternative sources and treatment practices, these regions are likely to face serious problems maintaining an adequate water supply [42]. For this reason, individuals’ perceptions of water scarcity seem to reinforce the population’s acceptance of alternative water treatments. Droughts appear to be a major influencing factor; the acceptance of alternative water treatment approaches is higher in regions experiencing water shortages [20,43,44].

Another variable that appears repeatedly in the literature relates to individuals’ knowledge about the technological and operational aspects of decentralized systems. In general, the users who are most informed about the technical reliability of the equipment, as well as aspects related to the operation and maintenance of these facilities, will tend to show a higher degree of acceptance [19,45,46]. However, as Ryan et al., report, careful targeting of the audience and the design of public communication campaigns relating to these systems are required in order to ensure successful communication [39].

(c)Perceived benefits 

Perceived benefits are another recurrent facilitator for the public acceptance of alternative wastewater systems. These benefits include savings on household water bills, delaying or eliminating the development of new water supply sources (i.e., eliminating the need to build new water treatment plants or desalination plants in favor of reusing treated wastewater), and mitigating the effects of water restrictions on lifestyles and property values [44,47,48,49]. 

Ho and Anda (2006) state that another perceived benefit of decentralized systems is ease of management [50]. Decentralized wastewater treatment systems are much easier to manage than traditional centralized plants, which are larger and need constant professional supervision. Several studies have confirmed this statement, both among actual system users [51] and potential users alike [52].

As one oft-repeated benefit in many studies, decentralized systems are perceived to be more sustainable and environmentally friendly [3,13,53,54]. Citizens believe that the use of decentralized systems increases their capacity and confidence to assist in and ensure environmental sustainability [55,56,57,58]. Feeling that they are part of a solution makes them more willing to accept alternative wastewater systems [59]. 

(d)Perceived costs

Many studies have clearly identified the costs of setting up and maintaining these decentralized systems as potential barriers to their adoption [19,41,48,51,60,61]. Additionally, the cost savings associated with alternative systems using recycled water take time to become apparent, which dampens customers’ enthusiasm [62]. Mankad and Tapsuwan (2011) suggest that willingness to adopt these systems could be increased if governments facilitate the adoption of decentralized systems, as well as the ease of public access to these forms of financial support [44]. 

Furthermore, in relation to the authorities, participants in some studies raised concerns about the legal framework needed to build and maintain the plants and facilitate the reuse of treated wastewater [48,54,63,64,65]. 

(e)Trust, social norms, and fairness

A growing number of studies have considered the importance of trust in those who are asking to introduce these new systems and the systems themselves [19,36,41]. This trust is related not only to citizens’ perceptions of the quality of the water originating in these alternative systems, but also those who promote the use of these alternative systems [13,47]. Public reluctance is related to mistrust of organizations that operate and manage these water systems and negative perceptions of those who own the infrastructure and their water supply decisions. Both factors were reported as being crucial issues that can affect the acceptance of alternative systems.

One must also consider social norms, which refer to things that a group approves of or are common practices among the group’s members. Social norms can be powerful influencing factors, affecting the attitudes and behavior of group members. According to this proposal, people are more accepting of decentralized systems if they perceive that important community members or close acquaintances support these systems. The respondents’ own level of support is also associated with their perceptions of how much the relevant others support other recycled water systems, such as decentralized plants [44,54,60,66].

Among the variables that have attracted less research attention but are nonetheless associated with the acceptance of decentralized systems is the perceived fairness of the recycled water scheme [67,68]. This variable is related to trust and can act as a shortcut for group members to decide whether they can trust an authority and accept its decisions. Being treated fairly indicates whether group members are respected and valued. People are often concerned about fair procedures (i.e., Who decided to implement the system and why? Are people involved in the decision-making process?) as well as fair outcomes (i.e., Who will pay the costs? Will this system create differences between groups? Will it affect water cuts or prices?). According to Nancarrow et al. (2010), where individuals perceive the system to be fair, their acceptance of the system increases [20]. In a study carried out by Mankad et al. (2010), concerns about fairness arose only when people were asked about preferences for community or individual water systems. Respondents indicated that they preferred individual water systems because of anticipated equity issues among the various users of the system [41].

(f)Factors related to the final use 

Health concerns are beyond the perceptions of risk [69]. They appear to be the most significant barrier to the increased acceptance of alternative wastewater treatment systems [50,55,70,71,72]. In general, citizens will be more accepting of decentralized systems if there are guarantees that public health will not be affected [73].

Health concerns have been shown to be related to the degree of personal contact with treated wastewater, regardless of the level of treatment. It is reasonable to suggest that public perceptions of risk associated with treated wastewater are linked to the use of reclaimed water from decentralized plants and their perception that it is “dirty” water. Several publications have convincingly and systematically illustrated that the acceptance of recycled water decreases when citizens’ physical contact with that water increases [37,74]. Acceptance is more likely when wastewater comes from a bath or shower, and the reuse of this treated water is for flushing the toilet or irrigation of gardens and streets [75]. Therefore, individuals’ level of comfort with water reuse is higher when the level of individuals’ physical contact with the reused product is lower; however, acceptance does depends on the final purpose for which the recycled product is used [47].

### 3.2. Barriers and Facilitators for Acceptance of Decentralized Wastewater Treatment Systems in Regions Where Citizens Are Unaware of Water Problems

The iterative process of collecting, coding, and analyzing the triangulated data resulted in a substantive theory composed of three central categories: (1) advantages; (2) disadvantages; and (3) application of decentralized wastewater treatment systems.

Figure 2 illustrates the content of the three axial codes that emerged from the focus group data. These axial codes are presented as diagrams, which comprise the open codes and their frequency of occurrence in the discourse (left number of each category) and their relationship with other codes (right number of each category). As for the central category of “advantages”, it can be seen that “support for the environment” (7–1), is mentioned seven times in this code, and “collective agreement” (6–2) appears six times in this category but is present in two codes (advantages and disadvantages). In the central category of “disadvantages”, the most-used argument was “conception of Galicia without water problems” (19–1). This argument was cited 19 times and appears only in this code. Finally, the central category of “application” (0–3) is related to three codes: “reluctance to use it as drinking water” (9–2), cost-benefit (5–1) and the acceptance of products derived from it (2–1). Figure 2 also reveals that, in relation to decentralized systems technology, focus group participants identified more disadvantages associated with implementation (in frequency and diversity) than expected benefits.

## 4. Discussion

Overall, several variables have proven to be crucial in explaining public acceptance of decentralized wastewater treatment in different contexts around the world. These variables could therefore be considered to be generalizable. In contrast, other variables seem to play a less clear-cut and possibly a moderating role, or appear only under certain circumstances. To organize the discussion below, the section follows the classifications derived from the literature review, highlighting which barriers and facilitating factors were also found in the region where the population was unaware of water issues such as water scarcity, drinking water quality and discharge of treated wastewater into water courses.

### 4.1. Factors Influencing Public Acceptance: Comparison between Worldwide Findings and Regions Where Citizens are Unaware of Water Problems 

(a)Sociodemographic variables

From the analysis of data on the public acceptance of decentralized wastewater treatment, acceptance levels are higher when interviewees are male, younger, and more educated. However, focus group participants did not mention the fact that any sociodemographic variable may affect the acceptance rates. Participants in a region where citizens are unaware of water issues did not spontaneously raise the idea that gender, age, or educational level might make a difference in whether someone accepts decentralized wastewater treatment plants.

(b)Perception of water scarcity, knowledge, and information

Several studies around the world have shown that when the perception of knowledge and information is high, the level of acceptance of decentralized wastewater treatment systems is also high. In particular, a perception of water scarcity is an important factor behind the public acceptance of alternative water treatment systems [20]. In the Galicia focus group, the most repeated element (19 times) was the perception that Galicia does not have water problems. This could serve as the main argument against the implementation of a decentralized system in the region. If decentralized systems are framed as a solution to problems caused by droughts, citizens are unlikely to accept the implementation of these systems in places where water scarcity is not a problem. These are relevant facts that will be examined in more detail later.

Participants in the focus group also mentioned “lack of knowledge” as a disadvantage for acceptance. Although focus group participants only mentioned this variable explicitly a couple of times, a closer look at the results reveals that its impact could be greater. Given that the interviewees had no experience with this decentralized system, it can be assumed that an argument such as “concern for change and cost” (15–1) is related to a lack of knowledge among participants. People tend to embrace concepts they are familiar with and reject those that they do not know about or understand, therefore this result also supports the finding in previous research that knowledge affects acceptance [44,55]. 

(c)Perceived benefits

As the literature review demonstrates, when individuals perceive benefits associated with decentralized wastewater treatment systems, such as saving water and money, avoiding water restrictions, and increasing environmental responsibility, they are more likely to accept these systems. 

In the Galicia focus groups, associating decentralized plants as systems that are easy to implement (9–1) and as systems imitating natural processes (4–1) were considered advantages. This element coincides with Ho and Anda’s (2006) work [50], which mentions this benefit as the perception of simple management. If decentralized plants can simplify the functioning of and maintenance associated with the wastewater treatment process, people will be willing to use them.

Focus group participants also believed that supporting the environment is an important advantage of the implementation of these systems (7–1). This result supports past findings, showing that people who felt a strong urge or obligation to protect the environment have more positive attitudes towards accepting alternative technologies [47]. 

Another perceived benefit that the focus groups identified, which did not appear in the literature review, is the generation of new employment opportunities (2–1). This is related to literature regarding the acceptance of new technologies [76]. If people perceive that the decentralized plants will create employment opportunities for the community, this will likely increase acceptance. 

The focus groups did not mention a decrease in the number of state-imposed water restrictions as a benefit. This was unsurprising, because water is not a scarce resource in Galicia. Therefore, it is understandable that participants in the focus group did not perceive this as a benefit of this new technology, although a reduction in water restrictions was an important perceived benefit in other contexts [61]. 

Moreover, focus groups did not mention the savings on household water bills, although this topic was discussed frequently in the studies from the literature review; decreased consumer costs were key to the acceptance of decentralized systems and the reuse of water in other contexts [77]. On the contrary, when participants in the focus groups referred to the economic side of this new technology, they mentioned perceived costs, not potential savings. 

(d)Perceived costs

As the literature review illustrates, perceived costs associated with installation, maintenance and compliance with government requirements is critical to the acceptance of these alternative wastewater treatments [41,78]. In addition, people often believe that decentralized water supply systems are more expensive than centralized ones, due to the misrepresentation of sunk costs and the lack of consideration of avoided costs [47]. The discussion groups in the instant study also came to these conclusions. Participants highlighted the short-term costs of the installation and implementation of decentralized systems (15–1), including the cost and amount of land needed (4–1), and expressed concern about the disuse of the current system (2–1).

Other perceived costs were related to energy consumption, general environmental impacts (9–1) and what is often termed the “Not in My Backyard” (NIMBY) effect [79,80,81]. The NIMBY effect refers to categories such as “keep waste away” (11–1), “concerns about plant location” (2–1) and “concerns about odor” (1–1).

This NIMBY effect did not explicitly appear in the systematic review. However, the effect is commonly seen with other projects, such as the construction of waste incinerators or waste plants near populated areas [82]. People are reluctant to have such a plant near their homes for fear of noxious side-effects, such as odors, gases, and aesthetic impacts, among others. In this case, the NIMBY effect is extrapolated to the implementation of decentralized plants, because centralized wastewater management systems are often built far from inhabited areas in Galicia.

Finally, another cost-related concern was the lack of support for implementation (1–1). In the literature, this is often referred to as legal arrangements [50]. To accept decentralized plants, the public must perceive that public administrations are willing to change the current funding system for large centralized hydroelectric projects, which to date have been subsidized by governments and international organizations [70].

(e)Trust, social norms, and fairness

As the literature review demonstrates, the acceptance of decentralized wastewater treatment systems is higher when citizens perceive the organizations that operate and manage the decentralized systems to be trustworthy, fair, and equitable. However, the discourse in the focus groups did not seem to address and was not clearly representative of any of these concepts. Instead, participants mentioned collective agreements. The fact that this concept was pointed out as an advantage (6–1) when there was a collective agreement, and as a disadvantage (6–1) when there was not, means that it plays an important role in this debate. Participants recognized that if citizens’ perceptions are ignored in implementation, this can sabotage the application of the new technology. The term ‘collective agreement’ is not mentioned in the literature, but it arguably has a similar meaning to the concept of commitment, which appears in some articles in the literature review [46,83,84]. Studies such as Mankad et al. (2010) conclude that the implementation of decentralized systems should involve conscious and active processes of engagement [41], ensuring that people are committed to a decentralized project from the beginning [59]. This demonstrates the importance of the process of social interaction and communication in shaping social frameworks and norms, especially in the implementation of decentralized systems where the public does not yet have a clear position on the technology.

(f)Factors related to the final use 

Health concerns related to the degree of physical contact people have with treated wastewater appears frequently in the literature. When the degree of physical human contact with wastewater is minimal, public health safety is ensured and the final use of treated water from decentralized plants is applied only for toilet flushing and garden irrigation, public acceptance increases. The comments of the participants in the focus groups echo this. Although the health risks themselves were not mentioned, concerns were raised about certain uses of reclaimed water. One of the most repeated comments concerned avoiding the use of reclaimed wastewater for drinking (9–1). This confirms that the public feels that the level of physical contact with reused water should be low, and acceptance also depends on the final use of the recycled product [47].

### 4.2. Fostering Public Acceptance in Regions Where the Public Is Unaware of Water Problems

Given the foregoing discussion, it is worth highlighting that in regions where people are unaware of water problems—which are critical for social acceptance of this technology—social acceptance is lower. The most important argument against decentralized systems (both in frequency and in relevance) that focus group participants mentioned was that Galicia has no overarching water management issues. This means that people perceive those plants as a possible solution for places suffering from drought or other environmental problems. Galicia does not experience such problems, therefore participants felt that it does not make sense to install decentralized plants.

People will be inclined to continue with the status quo routine where there are no underlying problems to prompt a change in approach. In this case, citizens felt it was unnecessary to use a decentralized wastewater treatment system given the lack of problems with drought in their community. However, participants also raised compounding arguments against the new decentralized technology, including uncertainty over the cost of implementation, concern about where plants would be built, and fears about the environmental impact of the projects. However, it might be possible to modify people’s perceptions about it.

An analysis of the results of the focus group reveals that participants also raised arguments in favor of decentralized plants. These arguments in favor were mainly related to lowering operating costs, the simplicity of implementing such a system, and supporting the environment. Environmental problems are a worrisome topic, which, according to Fernandes-Jesus, Lima and Sabucedo (2018) are becoming important in the personal and social identity of many people [85]. It is interesting to note that those arguments also coexist with their opposites in many people. When people argue, they often use rhetorical devices such as “on the one hand … but on the other hand” [86], which reflects their understanding that the positive and negative aspects of a given approach to an issue are integrally linked. Consequently, the possibility that people will change their minds when their beliefs are challenged with facts and arguments is more than wishful thinking.

To increase the acceptance of decentralized plants in regions with little awareness of water issues, messaging by proponents must highlight other qualities of this technology. For instance, proponents should discuss how the plants promote the more sustainable use of water resources and the circular economy concept. Moreover, it is important that proponents make these arguments in the context of an open exchange of ideas and debates among various stakeholders and social groups. By taking this approach, these arguments can be portrayed as being reflective of shared social beliefs, thus avoiding conflict and reactance [87,88]. These are the core ideas that proponents of decentralized plants must take into account when examining the framework of social acceptance. 

## 5. Conclusions

The aim of this study was to find out whether public perceptions of the implementation of decentralized plants in regions where people are unaware of water issues were the same as the barriers and facilitators found in similar research around the world. The findings showed that in these regions, the lack of awareness of water problems is critical to acceptance (or lack thereof). Therefore, public acceptance depends on how useful the public perceives these technologies to be, which should reinforce other qualities of this technology to overcome perceived barriers. 

Therefore, the question should not be whether a region is experiencing water scarcity or environmental issues. The problem is greater, and is related to whether there is a real commitment to a sustainable environment and a circular economy. This framework can lead to greater social acceptance, because it is related to values that are of increasing importance to the personal and social identities of many people. However, this does not guarantee acceptance, because both frameworks will compete. Therefore, citizens must reach a social agreement. In democratic societies, an issue as important as the use of water and water technologies requires an in-depth debate involving all stakeholders. Conducting fruitful public discourse will ensure that all stakeholders will feel co-responsible, and it will be easier to reach broad agreement.

This study examined the arguments both for and against decentralized plants in a region with specific characteristics (i.e., where there was little public awareness of water issues). Further research with the same focus group methodology in similar regions should be designed to determine whether a similar framework is active. On the other hand, based on the results obtained in this work, studies with representative samples could be designed to better understand the impact of these frameworks on the whole population, as well as on groups of special interest.

Moreover, future researchers could extrapolate the general conclusions of this research and apply it to other major challenges and transitions which society faces, such as climate change and the energy transition. Like the citizens of Galicia, who are largely unaware of issues pertaining to water, many citizens do not feel a sense of urgency or immediacy to address climate change. However, according to the United Nations, it is critical to take specific actions to mitigate environmental damage. Moreover, it is important to engage people in addressing these problems, by hearing their opinions and providing debate spaces (for instance, the focus groups discussed in this manuscript). This is helpful because actively thinking about a problem raises awareness of it. This sort of discussion also allows citizens to empower themselves, to decide to be part of the solution, and to propose specific actions to affect the change they want to see. The goal should not be to impose new technologies on citizens for the greater good in a top-down fashion, but to instead encourage bottom-up acceptance. As this research showed, citizens believe that decentralized wastewater treatment plants have many advantages and disadvantages. It is necessary to frame messaging to address those concerns and illustrate how the positive outcomes of these alternative approaches can create a more sustainable society.

## Figures and Tables

**Figure 1 ijerph-17-09060-f001:**
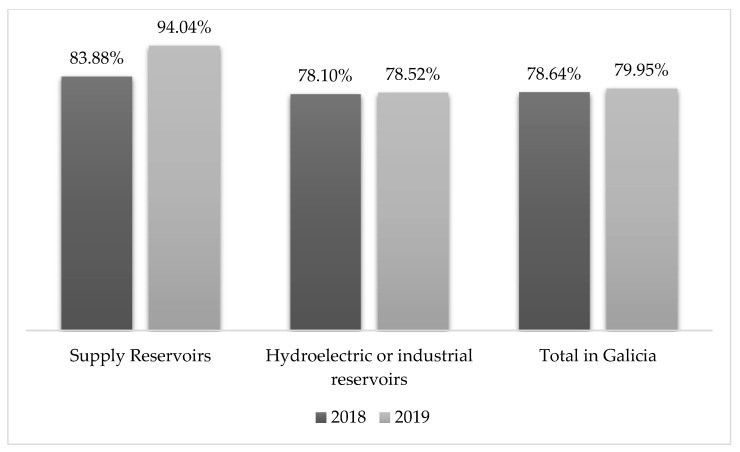
Comparison of the average occupation of the reservoirs in Galicia. Comparison between 2018 and 2019. Source: Hydrological bulletin of Xunta de Galicia [28].

**Figure 2 ijerph-17-09060-f002:**
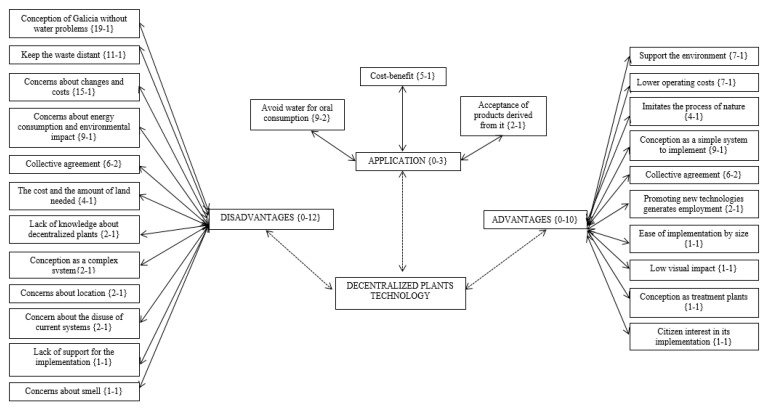
Summary table of the analysis carried out in Atlas.ti, which includes the recurring ideas from the focus groups. The table shows the triangulated data composed by three central categories: (1) advantages, (2) disadvantages and (3) application of decentralized wastewater treatment systems.

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
