# Peer review of "“Who Cares?”: The Acceptance of Decentralized Wastewater Systems in Regions without Water Problems"

_ijerph, 2020, doi:10.3390/ijerph17239060_

Round 1

Reviewer 1 Report

The authors investigated the acceptance of decentralized water systems in regions without water scarcity. Galicia was taken as focus area of study (case study). The research method is well described (use of focus groups) and the results are presented clearly and visualised in a diagram, representing perceived advantages and disadvantages including the ‘priority’ numbers of the arguments. Results are discussed in comparison to existing literature and conclusions are drawn. The article is well written and clearly structured. Method-wise I would like to express my compliments. 

Still I have a few additional questions which should be considered as suggestions.

First, I have a question on the method. If I look at the diagram in Figure 2 then I would be interested to learn more about the presented advantages and disadvantages and their priorities. There will be underlying causes which lead to this result. Also, the presented arguments (advantages and/or disadvantages) in Figure 2 could be (implicitly) linked with each other. Without asking the authors to provide these underlying causes and connections, I would be interested in a suggestion for a method/approach to find these causes and to add deeper layers to Figure 2.

Moreover, the research question of the authors is also important in wider and more generalized context. Each society is facing major challenges and transitions such as climate change, circular economy and the energy transition. Also in this context, a sense of urgency is currently not always felt collectively. However, timely change is critical to shield future generations and the environment from damage or disaster. This aspect of future risks (I am not an expert here but maybe water scarcity could become a problem in the future in Galicia) did not emerge from the focus groups. Could the authors possible elaborate (more) on potential important factors that may facilitate change, which do not automatically emerge from focus groups and how method-wise these unnoticed factors could be integrated in a study like this. Maybe if people were aware of future risks, they may have brought up other arguments.

In line with the previous suggestion, I would like to ask the authors to provide more elaboration on how the method and results benefit a more generalized context. How can this research (method) and results benefit other challenges such as climate change, a circular economy and energy transition? In other words, how can this approach facilitate change? Should we all start with focus groups to accommodate change processes? The answers I am looking for are touched in lines 487-489 and lines 502 - 508 but I believe more elaboration would strengthen the authors’ manuscript and increase its value for a wider audience.

Having said this, again I would like to express my compliments for the manuscript as it currently is.              

Reviewer 2 Report

In this manuscript, public perceptions of the implementation of decentralized plants in regions where people are unaware of water issues were compared with the barriers and facilitators found in similar research around the world. Reframing the usefulness of decentralized plants by focusing on environmental sustainability and circular economy, is key. This manuscript is an original and well-organized work. While this study would be a significant interest to the readers of Int. J. Environ. Res. Public Health, I recommend the publication of the paper after revision. More specific comments are as follow:

  1. Line 23: “……these conditions.”, which conditions?
  2. Line 28: “……the key..”, double full stops? Please check it through the text.
  3. Line 53: “(the product recovered also depends on the system;…….”, the parenthesis seems weird.
  4. Lines 213‒218: in this paragraph, the results of these papers including the specific literature of Mankad et al. (2011) should be provided or summarized.
  5. Lines 283‒290: this paragraph is repeated with the paragraph of Lines 275‒282.
  6. Line 515: “……. of special interest.6. Patents” It seems weird.
  7. The whole manuscript can be more concise and clear.
